# TSLP and HMGB1: Inflammatory Targets and Potential Biomarkers for Precision Medicine in Asthma and COPD

**DOI:** 10.3390/biomedicines11020437

**Published:** 2023-02-02

**Authors:** Fabiana Furci, Giuseppe Murdaca, Corrado Pelaia, Egidio Imbalzano, Girolamo Pelaia, Marco Caminati, Alessandro Allegra, Gianenrico Senna, Sebastiano Gangemi

**Affiliations:** 1Allergy Unit and Asthma Center, Verona University Hospital, 37134 Verona, Italy; 2Department of Clinical and Experimental Medicine, School and Operative Unit of Allergy and Clinical Immunology, University of Messina, 98125 Messina, Italy; 3Department of Internal Medicine, Ospedale Policlinico San Martino IRCCS, University of Genova, Viale Benedetto XV, n. 6, 16132 Genova, Italy; 4Department of Health Sciences, University “Magna Græcia” of Catanzaro, 88100 Catanzaro, Italy; 5Division of Internal Medicine, Department of Clinical and Experimental Medicine, University of Messina, 98125 Messina, Italy; 6Department of Medicine, University of Verona and Verona University Hospital, 37134 Verona, Italy; 7Division of Hematology, Department of Human Pathology in Adulthood and Childhood “Gaetano Barresi”, University of Messina, 98125 Messina, Italy

**Keywords:** TSLP, HMGB1, COPD, asthma, alarmins, airway inflammation

## Abstract

The airway epithelium, through pattern recognition receptors expressed transmembrane or intracellularly, acts as a first line of defense for the lungs against many environmental triggers. It is involved in the release of alarmin cytokines, which are important mediators of inflammation, with receptors widely expressed in structural cells as well as innate and adaptive immune cells. Knowledge of the role of epithelial cells in orchestrating the immune response and mediating the clearance of invading pathogens and dead/damaged cells to facilitate resolution of inflammation is necessary to understand how, in many chronic lung diseases, there is a persistent inflammatory response that becomes the basis of underlying pathogenesis. This review will focus on the role of pulmonary epithelial cells and of airway epithelial cell alarmins, in particular thymic stromal lymphopoietin (TSLP) and high mobility group box 1 (HMGB1), as key mediators in driving the inflammation of chronic lung diseases, such as asthma and chronic obstructive pulmonary disease (COPD), evaluating the similarities and differences. Moreover, emerging concepts regarding the therapeutic role of molecules that act on airway epithelial cell alarmins will be explored for a precision medicine approach in the context of pulmonary diseases, thus allowing the use of these molecules as possible predictive biomarkers of clinical and biological response.

## 1. Introduction

The airway epithelium, the first line of defense against environmental triggers, such as allergens, viruses, pollutants, and microbes, is assuming an increasingly interesting role both regarding knowledge of pathogenesis and the current and possible therapeutic strategies of chronic inflammatory lung diseases. Airway epithelial cells express pattern recognition receptors (PRRs), transmembrane or intracellularly, that are able to recognize highly conserved microbial motifs called pathogen-associated molecular patterns (PAMPs) and host-derived molecular damage-associated molecular patterns (DAMPs) [1,2,3]. PRRs play a key role in recognizing invading threats and in initiating host defense through the rapid production of alarmin cytokines [4,5]. DAMPs released from infected or damaged cells bind to specific DAMP receptors (DAMPRs) that are able to stimulate cell differentiation, cell death, or the secretion of inflammatory or anti-inflammatory cytokines.

In reality, the pulmonary epithelium is complex both functionally and structurally, and is a tightly regulated system. In particular, the various tracts of the respiratory airways are made up of diverse cell types, characterized by an organized network of cell–cell interactions and regulatory mechanisms which are fundamental for maintaining tissue homeostasis and responding effectively to damage [6,7]. Proximal airways consist of specialized epithelial cell populations ranging from simple ciliated cuboidal cells to pseudostratified ciliated, columnar cells, while the distal alveolar region consists of alveolar epithelial cells (AECs) [5]. AECs can be further classified into type I or type II pneumocytes, which are different from a phenotypic and functional point of view. In particular, type I pneumocytes (AECI), which make up about 90–95% of the alveolar surface, play a key role in gas exchange, while type II pneumocytes (AECII) are specialized in the production and secretion of pulmonary surfactants, which are fundamental in preventing alveolar collapse [7,8]. In the lung, epithelial cells localized at the interface between the internal and external environments, carrying out a key role in maintaining tissue integrity such as mucociliary clearance, tight junctions, epithelial adherence, and secretion of antibacterial, antimicrobial, and antiprotease molecules [9,10,11,12,13,14], are exposed to many exogenous stressors, for instance bacterial and viral insult, cigarette smoke, asbestos, and airborne pollutants [15], and many endogenous signals such as oxidative stress [16].

Epithelial cells are essential to regulate immune response and facilitate the resolution of inflammation, restoring tissue homeostasis in response to injury [17], albeit in many chronic lung diseases persistent inflammation, pathological repair mechanisms, and functional tissue architecture destruction may arise [18], where injury to epithelial cells underlies the pathogenesis of many chronic pulmonary diseases [19].

Chronic airway diseases, for instance asthma and chronic obstructive pulmonary disease (COPD), are characterized by various inflammatory phenotypes and an underlying abnormal immune response. These two inflammatory respiratory diseases have similar symptoms, such as dyspnea, cough, and sputum, but differ in pathogenesis, age of onset (young age for asthma, old age for COPD), disease progression, prognosis, and therapeutic strategies. In asthma, the main cells involved are CD4, T-lymphocytes, eosinophils, and mast cells; in COPD, the main cells involved are CD8, T-lymphocytes, and macrophages. In severe cases of both chronic respiratory diseases, there may be an infiltration of neutrophils in the airway walls. The thickening of smooth muscles is typical in large airways in severe asthma and in small airways in COPD. In COPD, there are typical structural alterations, such as epithelial changes, caused by cell metaplasia, mucous membrane changes, and fibrosis of airway walls. Moreover, another feature typical of COPD but not of asthma is the destruction and fibrosis of the alveolar wall [20].

Allergens and pathogens, as well as smoking and further pollutants, act on the lung inducing injury and airway inflammation in a context of genetic predisposition and altered immunity, leading to fixed airflow obstruction and subsequent characteristic symptoms of COPD [19,21]. In COPD, which has a high prevalence and incidence of morbidity and mortality worldwide, a major characteristic of the disease is airway inflammation and this is involved in both progression and pathogenesis, although anti-inflammatory treatment may not be a first-line therapy. In COPD, inflammation can be seen in both the small and large airways and can persist after smoking cessation, probably due to the presence of modified immunity and alterations in the airway microenvironment [20,22].

In the most common phenotypes of COPD, both innate and adaptive immunity are involved, with neutrophilic inflammation. Cigarette smoke exposure, along with other pollutants and allergens, cause airway damage with a consequent release of pro-inflammatory mediators and DAMPs, such as interleukin (IL)-33, thymic stromal lymphopoietin (TSLP), and high mobility group box 1 (HMGB1) [19]. DAMPs, members of the alarmins, which are warning signals released from damaged cells, alert the immune system by activating inflammasomes through interaction with PRRs localized on the plasma membrane, inside endosomes after endocytosis, in the cytosol receptor for advanced glycation end products (RAGE), on retinoic acid-inducible gene 1(RIG-I)-like receptors and nucleotide-binding oligomerization domain 1 (NOD1)-like receptors, and absent in melanoma 2 (AIM2)-like receptors [19].

IL-33 receptor ST2 distribution is modified after exposure to cigarette smoke: it is downregulated in innate lymphoid type-2 cells (ILC2) and upregulated by macrophages that leads to triggering of an IL-33-dependent exaggerated pro-inflammatory cascade [5]. Airway injury alters the barrier function, predisposing the airway to bacterial dysbiosis and infection, which, along with pollutants, induce the switch of ILC2 cells to ILC1 cells, intensifying type-1 inflammation [6,7,19]. While epithelial cells are implicated in inflammatory mediator release, such as tumor necrosis factor (TNF), IL-1β, IL-6, and IL-8, macrophages are involved in releasing pro-inflammatory cytokines and activating the NLRP3 inflammasome with caspase-1-dependent release of IL-1α, IL-1β, IL-33, and IL-18 [23].

Although in COPD the neutrophil phenotype is the most frequent finding, 10–40% of patients with COPD report elevated eosinophilic inflammation in the sputum and/or blood, whose etiology is currently unknown. This eosinophilic inflammation, as in asthma, is related to the risk of severe exacerbations and increased T2-transcriptome signatures.

Similar to neutrophil-associated COPD, eosinophilic COPD includes activation of both innate and adaptive immunity [24,25]. As in asthma, allergies promote Th2 cells that produce IL-4, IL-5, and IL-13. However, eosinophilic inflammation may also be activated by ILC2 cells producing IL-5 and IL-13 in response to PGD2 and the epithelial-derived alarmins IL-33, IL-25, and TSLP released after epithelial injury caused by microbes and pollutants. A further role may be mediated by macrophage-derived IL-33, released after inflammasome activation [24].

Therefore, an important aspect to highlight is that airway inflammation plays a key function in asthma and COPD development, but it is also important to consider that while in COPD there is increased lung inflammation that causes severe damage to the lung parenchyma, in asthma we see local chronic airway inflammation that causes airway remodeling and hyper-responsiveness. In both pathologies, there is the involvement of the adaptive and innate immune system [26]. Injured or dying cells release DAMPs which can warn the immune system through interacting PRRs. DAMPs may be categorized according to their origin. In particular, DAMPs comprise many nuclear proteins, for instance IL-1a, IL-33, and HMGB1, cellular organelles, for example, histones and mitochondria and other cellular components, for instance uric acid, adenosine triphosphate (ATP), heme, HSPs, defensins, haptoglobin, EDN, LL-37, galectin-3 (Gal-3), S100 proteins, hyaluronan, and heparan sulfate [27]. Elevated levels of many DAMPs, including HMGB1, were reported in induced serum and sputum of patients with asthma and COPD in many studies [5,6,19,27,28].

Therefore, in this paper we aim to analyze the role of alarmins, in particular of HMGB1 and TSLP, in asthma and COPD, identifying similarities and differences which may be useful for therapeutic approaches that are as targeted as possible.

## 2. HMGB1: Comparison in Asthma and in COPD

HMGB1, the most frequently expressed of the entire HMG family proteins, is a proinflammatory mediator that belongs to the alarmin family, playing a significant role in various acute and chronic immune disorders as a signal of tissue injury and a mediator of inflammation, whose levels are increased in many inflammatory conditions, such as sepsis, cystic fibrosis, and rheumatoid arthritis [29,30]. In particular, HMGB, a non-histone and ubiquitous chromosomal protein found enriched in active chromatin forming part of the high mobility group family of proteins, is encoded by the HMGB1 gene (13q12) in humans and has precise motifs which are DNA-binding domains. In particular, there are four categories of HMGBs (HMGB1-4) [31,32,33].

Many studies have reported an active extracellular release of HMGB1 from cells of the immune system or passively released from injured cells [21,22]. Moreover, studies have described the important role of HMGB1 in regulating several receptors that, through interaction with RAGE and Toll-like receptors (TLRs), are implicated in phosphorylation and in the synthesis of glycation end products. TLRs induce a growth of some cytokines, such as IL-4, IL-6, and TNF-alpha, typical of the innate inflammatory response. Therefore, increased levels of HMGB1 have been associated with several inflammatory diseases [19,28]. Watanabe et al. first tested the HMGB1 level and levels of the endogenous secretory RAGE (esRAGE) in the sputum of 44 asthmatic patients (previous to asthma treatment) and 15 controls, highlighting that sputum levels were notably increased in patients with asthma compared to controls, with a positive correlation between HMGB-1 level and disease severity, while for esRAGE there was no significant difference between mild persistent and severe asthmatic patients [34]. Ferhani et al. described levels of HMGB1 in the fluid from bronchoalveolar lavage (BAL) as also being high in patients with COPD [35]. Straub et al. showed that HMGB1 inhibitors undergo a significant reduction in the ovalbumin-induced rise in response to methacholine in a mouse asthmatic model sensitized and challenged with ovalbumin [36]. Hou C. et al. reported that plasma and sputum concentrations of HMGB1 in patients with asthma and COPD were significantly more elevated than concentrations in control subjects and were significantly negatively correlated with forced expiratory volume in 1 s (FEV1), FEV1 (% predicted). Levels of HMGB1 in the induced sputum of patients with COPD were significantly higher than those of patients with asthma and healthy controls not reporting significant differences in HMGB1 levels between patients with eosinophilic and non-eosinophilic asthma [37]. This latter aspect can be explained by the different sources of HMGB1, such as macrophages, eosinophils, and neutrophils. HMGB1 levels in patients with asthma and COPD reported a positive correlation with neutrophil counts and percentage of neutrophils [38,39]. Hou C. et al. reported also that HMGB1 in induced sputum correlated positively with the percentage of neutrophils and neutrophil count: this is backed up by the finding that neutrophils contribute to higher levels of HMGB1 and that purified HMGB1 may stimulate the dose-dependent chemotaxis of neutrophils [37]. In the literature, it has also been reported that higher numbers of alveolar macrophages can augment levels of HMGB1 in BAL fluid (BALF) from smokers with COPD, and these data are in accordance with Hou C. et al., who reported that macrophage numbers correlated positively with levels of HMGB1 in the sputum of patients with COPD [39,40]. Ferhani et al. also described bronchial epithelial cells as being a potential source of HMGB1 in the airways of patients with COPD, which is in agreement with other studies that reported that hydrogen peroxide may trigger the expression and release of HMGB1 from bronchial epithelial cells in vitro [35,41,42]. Liang et al., having isolated normal human bronchial epithelial cells from the human lung tissue of four patients undergoing lobectomy, highlighted that HMGB1 induces an augmented expression and secretion of TNF-a, TSLP, MMP-9, and VEGF and stimulates activities of the p38 MAPK and ERK1/2 pathways in bronchial epithelial cells by an enhancement of TNF-a, VEGF, MMP-9, and TSLP levels [43]. Cuppari et al. conducted a study on 50 children-adolescents affected by severe, moderate, and mild asthma and 44 healthy controls, reporting a significant increase of sputum HMGB1 levels in patients with asthma compared to healthy controls. Moreover, patients with severe asthma showed higher levels of sputum HMGB1 than patients with mild and moderate asthma. Furthermore, the authors highlighted a positive correlation between total serum IgE levels in the asthmatic group and sputum HMGB1 values in asthmatic children, while reporting an inverse correlation between levels of sputum HMGB1 and indices of lung function [29]. Ojo et al., in 2015, for the first time reported the role of HMGB1 in promoting bronchial epithelial cell wound repair, a process mediated by the production of integrins and ECM proteins [44]. Shang et al. first reported increased levels of plasmatic HMGB1 in patients with COPD compared to healthy subjects [45]. This result was confirmed by other studies such as Ferhani et al. who evaluated HMGB1 as an inflammation marker and analyzed the BAL of 20 never-smoker patients, 20 smokers, and 30 smokers with COPD. Bronchial biopsy and lung tissue sections were used to assess the immunolocalization of HMGB1 and RAGE. The authors observed that BAL levels of HMGB1 were similar in non-smoker healthy patients and in smokers without COPD but there were increased levels in smokers with COPD. A positive correlation between BAL levels of HMGB1 and TNF-RII and IL-1b, and a negative correlation with IL-1RA with no correlation with TNF-α was also reported. High concentrations of HMGB1-positive cells were seen in the bronchial mucosa of smokers with COPD more frequently than in healthy smokers. Lastly, the authors reported an overexpression of RAGE in the smooth muscle and airway epithelium of patients with COPD and that it colocalized with HMGB1 [35]. Pouwels et al. studied HMGB1 levels in the serum of 40 patients with COPD during both an exacerbation and stable phase. An equivalent study was conducted on the inducted sputum of 35 patients reporting no significative differences in levels of sputum, but with increased serum levels of HMGB1 during COPD exacerbations. A gender difference was, moreover, highlighted; in particular, there were high HMGB1 serum levels in women during the acute phase [46]. Cheng et al. reported upregulated TLR4 expression in the lung tissues of mice exposed to cigarette smoke and upregulated HMGB1 levels in lung tissues and airways due to extracellular translocation from the cytoplasm. However, inhibition of HMGB1 expression was related to reduced airway inflammation in mice. Exogenous HMGB1 was related to an increased proinflammatory cytokine production in wild-type mice, but it did not affect proinflammatory cytokine production in TLR4-KO mice [47].

## 3. Role of Thymic Stromal Lymphopoietin (TSLP) in Asthma and COPD

TSLP, a four α-helical type I cytokine, has effects on neutrophils, mast cells, basophils, eosinophils, ILC2s, natural killer T cells, smooth muscle cells, and tumor cells [48,49]. The primary sources of this cytokine are epithelial cells and stromal cells in the lungs, skin, and gastrointestinal tract during both homeostatic and inflammatory conditions, although dendritic cells (DCs), basophils, and mast cells can also produce this cytokine [50,51]. The production of TSLP in the lungs is also induced by viral infections, such as respiratory syncytial virus (RSV), rhinovirus, influenza virus, and lymphocytic choriomeningitis virus [52,53]. TSLP acts as an alarmin, indeed it is released from cells rapidly and incites further exogenous and endogenous danger signals, exacerbating inflammation. Its production is positively mediated by proinflammatory TH2-type cytokines, such as IL-4 and IL-13, by TNF, IL-1β, and IL-25. By contrast, its release is inhibited by interferon-γ (IFNγ) and IL-17 26, β2-Adrenoceptor agonists, and glucocorticoids [54]. An important aspect to highlight is the role of TSLP in allergic conditions that can be included considering that the cross-linking of IgE bound to the high-affinity receptor (FcεRI) stimulates mast cell production of TSLP [50]. Two variants of human TSLP have been investigated: a short form (sfTSLP) and a long form (lfTSLP) with a distinct regulation that implies different roles, with an anti-inflammatory or antibacterial function proposed for sfTSLP and a pro-inflammatory function for lfTSLP [55,56]. In relation to this concept, it is reported that in a mouse asthma model, the use of sfTSLP improved house-dust-mite-induced asthma, whereas the use of lfTSLP induced damage to airway barrier function [57]. To understand why single nucleotide polymorphisms (SNPs) rs2289276 and rs2289278 in the TSLP promoter are linked to a rise in childhood atopic disease and adult asthma, it is important to report that these SNPs promote increased binding by AP1 and increased lfTSLP production [58]. Certain TSLP gene variants have been identified, as outcomes of genome-wide association studies, in individuals with asthma, in particular with the association of TSLP SNP rs1837253 with childhood-onset asthma [59,60]. In mouse models, overexpression of TSLP in the lungs is related to airway hyper-responsiveness (AHR) and severe airway inflammation. Patients with asthma, in particular patients with severe asthma, show elevated levels of TSLP and TH2 cytokines in the airways [61]. A positive relation between TSLP levels and the risk of future asthma exacerbation has also been reported [62]. From biopsy sections of mild atopic patients with asthma, it has been seen that allergen challenge induced an increase in IL-25, IL-33, and TSLP levels in the bronchial epithelium and submucosa, with a positive correlation between these cytokines and the extent of airway obstruction [63]. In patients with eosinophilic asthma, increased levels of IL-4 were reported, responsible for the increased permeability of airway epithelial cells by decreasing expression of adhesion molecules and filaggrin, comprising E-cadherin, and, moreover, increasing IL-33 and TSLP levels, which further boosts TH2 inflammation [64]. However, while asthma is usually associated with eosinophils, TH2 cells, and/or mast cells, and COPD is commonly linked to TH1 cells, neutrophils, and macrophages, in COPD protein levels and TSLP mRNA were elevated in the bronchial epithelium compared with the controls [65]. These data are related to the correlation between the known factors responsible for COPD exacerbations, such as respiratory viruses, cigarette smoke, and pro-inflammatory cytokines, and TSLP production in COPD patients [53,66]. This helps in understanding the impact of TSLP in the development and/or exacerbation of COPD, and in what way TSLP affects lung pathophysiology in situations other than TH2-related asthma [54].

## 4. HMGB1: Epigenetics and Omics Approaches

HMGB1, whose biological functions are related to its subcellular localization and expression, plays a key role in the nucleus and cytoplasm as a DNA chaperone, chromosome gatekeeper, autophagy maintainer, and protector from apoptotic cell death, with a key role as an extracellular alarmin. Indeed, as reported in the literature, HMGB1 is a sophisticated signal of danger, which is becoming an increasingly useful clinical biomarker for several disorders. In physiological conditions, HMGB1, which is localized in the nucleus, moves to the cytosol after cellular exposure to cell stress responses and inflammation. Therefore, HMGB1, expressed in the cytosol, nucleus, or extracellular space, is released passively by lytic cell death or secreted actively by viable cells. HMGB1 shuttles actively between the nucleus and cytoplasm and plays a key role in modulating cell stress responses and inflammation [67]. HMGB1 stimulates DCs and macrophages to release inflammatory cytokines and TNF-α and chemokines through the TLR4/MD2/MyD88/NFκB pathway. Activated macrophages, which are involved in the secretion of HMGB1 in extracellular space, express receptors (RAGE, TLR-2/4) on their membrane that HMGB1 binds to [68]. Via RAGE, HMGB1 activates caspase-1 and inflammasome in macrophages, resulting in pyroptosis [48,49]. The internalization of RAGE/HMGB1 in macrophages induces the rupture of lysosomes and the release of cathepsins into the cytosol, stimulating inflammasome activation [67]. An important aspect to consider is the proposal that regulated secretion of HMGB1 takes place through its packaging into intracellular vesicles, such as autophagosomes or lysosomes, which release HMGB1 after fusion with the plasma membrane [69]. In the literature, the presence of HMGB1 in extracellular vesicles (EVs) has been reported, and nanoscale lipidic particles are released from all cytotypes, which comprise lipids, proteins and nucleic acids, in pathological and physiological conditions [70,71,72,73,74]. EVs may be gathered by many biofluids and have a fundamental role in health and pathological conditions, playing a relevant role in the modulation of intercellular communication [75,76]. In vitro and in vivo studies have reported that, in order to regulate vascular homoeostasis, endothelial cells (ECs) and endothelial-vascular smooth muscle cells (VSMC) release EVs containing HMGB1. The role of EVs is related to their cells of origin and this aspect is important in understanding how HMGB1 within EVs released by gastric cancer cells causes autophagy and pro-tumor activation of neutrophils via the HMGB1/TLR4/NF-κB axis; EVs containing HMGB1 released by bone marrow mesenchymal stem cells (BMSC) are modified, weakening the injury provoked by smoke inhalation. Indeed, smoke can stimulate expression of HMGB1, NF-κB, and inflammatory and apoptotic factors. BMSC-EVs significantly inhibited apoptotic factors, including cleaved caspase-3, Bax, and c-Jun, and reversed HMGB1 increase [75,77]. Moreover, epigenetic mechanisms, for instance the activities of microRNAs (miRNAs), may play an important role in regulating HMGB1 expression. Indeed, HGB1 can curb the expression of several miRNAs, a family of small, noncoding RNAs of 18–22 nucleotides that regulate gene expression associated with oxidative-stress-related chronic diseases [78].

It has been reported that many miRNAs modulate HMGB1 expression and its functions also in pulmonary disorders. In particular, it is reported that MiR-627 targeting HMGB1 modulates TGF-β causing pulmonary fibrosis via the NF-κB regulatory loop; in targeting HMGB1, miR-574-5p weakens acute respiratory distress syndrome and, in controlling fundamental aspects of respiratory system homeostasis, is a negative regulator of inflammatory responses [79,80]. Therefore, these data can be useful for further investigations and studies, both regarding epigenetics and omics approaches, with the chance to use genetic therapies or therapies involving EVs in future, potential cell-free strategies for lung damage and for monitoring the course of oxidative chronic diseases [81,82].

## 5. Summary and Discussion

This review summarizes the current literature on the role of pulmonary epithelial cells and DAMPs, in particular TSLP and HMGB1, in chronic lung diseases, such as asthma and COPD. DAMPs, members of the alarmins family, secreted from permanently damaged cells, have the role of alerting the immune system by activating inflammasomes through interaction with PRRs [19,28,46]. Precisely because of the above-described role of TSLP and HMGB1 in modulating inflammation in respiratory pathologies, from a careful analysis of the literature it can be seen that these molecules could increasingly represent therapeutic targets and biomarkers of response to both commercial and future therapeutic treatments.

TSLP, an alarmin cytokine that is released from airway epithelial cells exposed to environmental triggers, plays a key role in the pathobiology and pathophysiology of chronic respiratory diseases, such as COPD and asthma. In particular, this role can be understood if we consider how TSLP drives airway inflammation through the release of T2 cytokines, becoming a new pharmacological target to block T2 pathways involved in asthma. A block of the function of TSLP induces an inhibition of allergen-induced responses, such as bronchoconstriction, airway hyperresponsiveness, and inflammation [5].

Many studies have focused on the role of HMGB1, which is actively released extracellularly from cells belonging to the immune system or passively released from damaged cells, and which, in addition to its nuclear functions, has extracellular activities [28,83,84]. HMGB-1 has many functions, both in the nucleus, such as DNA replication, repair, recombination, transcription, apoptosis, and genomic stability, and outside the cell, as a signal for cell growth, proliferation, and death [19,28]. What can be appraised from this review of the literature was the verification of alarmin, in particular HMGB-1 and TSLP, contributing to chronic inflammatory pathologies.

Many studies demonstrated increased levels of HMGB-1 in diverse tissues of smokers and patients with COPD and higher HMGB-1 sputum levels in patients with severe asthma compared to moderate and mild asthma patients [19]. The studies reviewed help us to understand in what way this alarmin is associated with the development of COPD, in which cigarette smoking is the most well-known causative factor, inducing neutrophil death and necrosis. Necrosis of neutrophil cells induces DAMP release, in particular HMGB1, which is responsible for the recruitment of other neutrophils. DAMPs have a key function in triggering the inflammatory process by interaction with RAGE and in the worsening of COPD exacerbations. It was reported that the blocking of NF-*κ*B leads to an under expression of HMGB1 in lung tissue [28]. Animal studies confirmed an improvement in the condition of animals and a decrease in in situ inflammatory markers after the administration of drugs known to lower HMGB-1 levels or of anti-HMGB-1 antibodies to asthma subjects. Furthermore, many genetic variants for TSLP are associated with disease severity, and chromosomal alterations in TSLP are frequent in some cancers, highlighting the relevant role of TSLP in disease. The production of TSLP is caused by many environmental agents, such as ligands for TLRs, mechanical injury, cytokines, and viruses. Epithelial cells are the main source of TSLP production, which has pleiotropic actions on B cells, T cells, eosinophils, ILC2s, NKT cells, macrophages, smooth muscle cells, basophils, and mast cells. TSLP promotes and amplifies type-2 immunity, and this fosters the immune response to allergens or antigens through both innate and adaptive immune pathways, inducing the development and progression of allergic disease. Increased levels of TSLP and Th2 cytokines in the airways are associated with the risk of asthma exacerbation [61,62]. Biopsy sections from patients affected by mild atopic asthma have shown that allergen challenge raises IL-25, IL-33, and TSLP levels in the submucosa and bronchial epithelium, with a correlation between these cytokine levels and the degree of airway obstruction [63]. Moreover, although in COPD there is a predominant TH1 inflammation, increased protein levels and TSLP mRNA are reported in the bronchial epithelium of COPD compared with controls, suggesting the involvement of this alarmin in the development and/or exacerbation of COPD [54,65]. Tezepelumab, a human monoclonal antibody (IgG2λ) that binds specifically to TSLP, inhibiting interactions with its heterodimeric receptor, is hypothesized as representing a novel approach in the treatment of several phenotypes and endotypes of asthma. Tezepelumab is well tolerated and safe, and induces improvements in asthma control, reducing the frequency of exacerbations and hospitalizations in patients with severe asthma, regardless of phenotype or endotype. Clinical benefits are associated with reductions in levels of cytokines, such as IL-5 and IL-13, and in the number of airway eosinophils, independently of patient baseline biomarker status, suggesting a cellular mechanism based on the hindering of signals upstream of eosinophil activation [85]. The possibility of an anti-TSLP treatment allows us to understand how alarmins can represent not only possible “signals of inflammation” but also new therapeutic strategies of clinical precision medicine to deliver clinically meaningful improvements in patients with chronic airway inflammation. Therefore, the data we analyzed suggest a potential role of these alarmins in the classification of severity of respiratory chronic diseases and their possible use as a predictive marker of therapeutic efficacy and disease severity (Figure 1). Indeed, their use as biomarkers of therapeutic efficacy and disease severity could help clinicians in the choice of more specific personalized therapy for each patient affected by respiratory inflammatory diseases. In this way, it may be possible to target the inflammatory cascade which alarmins activate in the course of the disease. Moreover, these clinical, diagnostic, and therapeutic strategies may lead to new studies being drawn up to identify future therapies.

## Figures and Tables

**Figure 1 biomedicines-11-00437-f001:**
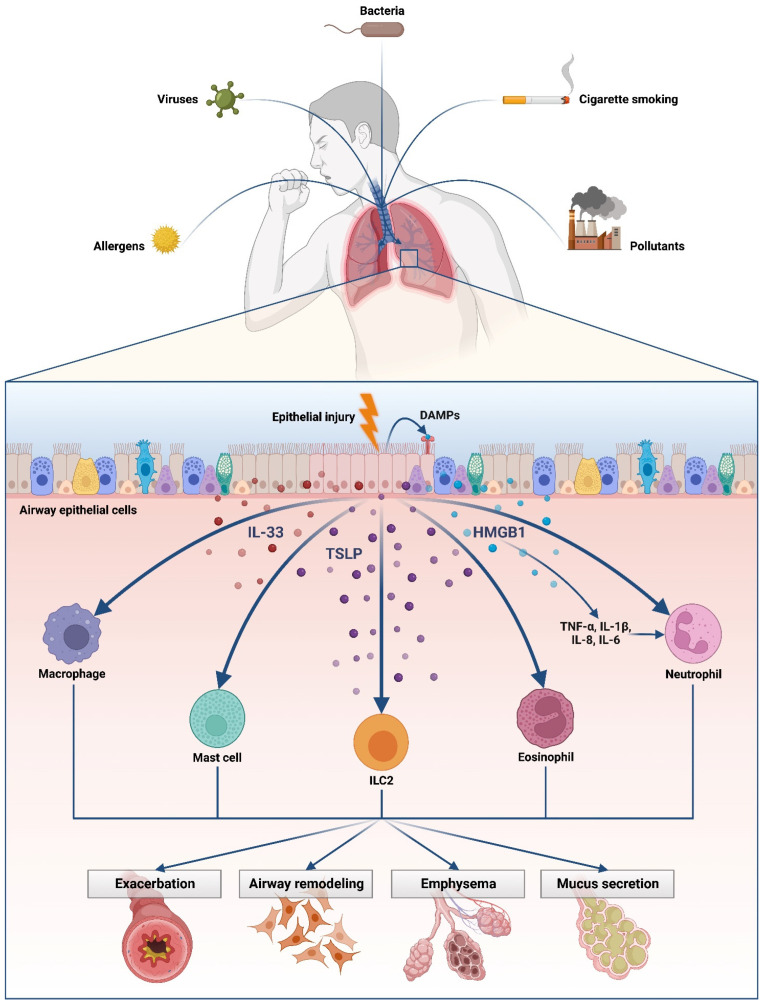
Description of molecular implications of airway epithelial cell alarmins in the pathogenesis of chronic inflammatory lung diseases, in particular in the pathogenetic mechanism that induces exacerbations, airway remodeling, emphysema, and mucus secretion.

## Data Availability

Data are contained within the article.

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
