# Peer review of "TSLP and HMGB1: Inflammatory Targets and Potential Biomarkers for Precision Medicine in Asthma and COPD"

_biomedicines, 2023, doi:10.3390/biomedicines11020437_

Round 1

Reviewer 1 Report

I have the article by Furci et al. with great interest. The authors summarised the role of TSLP and HMGB1 in asthma and COPD.

Comments:

·       Most importantly, the clinical message of the manuscript is not clear. The authors need to emphasise throughout the text if higher levels of TSLP and HMGB1 are associated with different phenotype (i.e. frequent exacerbations, more progressive disease, responsiveness to therapy).

·       Abstract. Please, avoid abbreviations, and write out the full name of the two molecules.

·       Page 2. Line 18. Delete “damage” in the end of the line.

·       Page 2. 3rd paragraph. Please, describe asthma and COPD better by focusing on similarities and differences in the pathology and clinical features.

·       Page 3. You may consider citing https://pubmed.ncbi.nlm.nih.gov/33546498/ which is a comprehensive review of eosinophilic COPD.

·       Page 3. Lines 7-13. There are a few facts, but citations justifying them are missing. Please, add references.

·       Page 3. Line 50. Please, delete “normal” before “controls”.

·       Instead of “asthma patients” and “COPD patients” please, use “patients with asthma and COPD”.

·       English is fine but not good. Some sentences are unnecessarily redundant and complicated. Please, ask a native speaker to check the paper and acknowledge them in the manuscript.

Author Response

Dear reviewer, I have revised the paper accordingly to your suggestions

Giuseppe Murdaca

Reviewer 2 Report

An important and complex article for the best knowledge of the pathophysiology of asthma and copd. This is a review article on a topic of high clinical and scientific interest.

Author Response

(The authors gave the same response as above.)

Reviewer 3 Report

1) Abstract. Moreover, emerging concepts around the therapeutic role of molecules  that act on airway epithelial cell alarmins will be explored for an approach of precision medicine in  the context of pulmonary diseases. Please underline the novelty of this manuscript.

2) 1. Introduction L34-40. The airway epithelium, the first line of defense against environmental triggers such  as allergens, viruses, pollutants, and microbes, is assuming an increasingly interesting  role both regarding knowledge of pathogenesis and current and possible therapeutic  strategies of chronic inflammatory lung diseases. Airway epithelial cells express pattern  recognition receptors (PRRs), transmembrane or intracellularly, that are able to recognize  highly conserved microbial motifs called pathogen-associated molecular patterns  (PAMPs) and host-derived molecular damage-associated molecular patterns (DAMPs). In order to discuss the previously described points, important references are needed to be added, such as:

a-Systemic and Airway Epigenetic Disruptions Are Associated with Health Status in COPD. Biomedicines 202311, 134. https://doi.org/10.3390/biomedicines11010134

b-The History and Mystery of Alveolar Epithelial Type II Cells: Focus on Their Physiologic and Pathologic Role in Lung. Int J Mol Sci. 2021;22(5):2566. Published 2021 Mar 4. doi:10.3390/ijms22052566

c- Alveolar Epithelial Type II Cells. Encyclopedia of Respiratory Medicine (Second Edition),2022, 10-17, 2022. DOI:10.1016/B978-0-08-102723-3.00157-8.

3) 1. Introduction. Albeit in COPD neutrophil-phenotype is the most frequent finding, 10–40% of pa- tients with COPD report elevated eosinophilic inflammation in the sputum and/or blood, whose etiology in currently unknown. This eosinophilic inflammation, as in asthma, is  related with the risk of severe exacerbations and increased T2-transcriptome signatures. Similar to neutrophil-associated COPD, eosinophilic COPD includes activation of  both innate and adaptive immunity.  Please, improve this paragraph and add several references:

a- COPD Definition: Is It Time to Incorporate Also the Concept of Lung Regeneration's Failure? [published online ahead of print, 2022 Sep 29]. Am J Respir Crit Care Med. 2022;10.1164/rccm.202208-1508LE. doi:10.1164/rccm.202208-1508LE

b- Definition and Nomenclature of Chronic Obstructive Pulmonary Disease: Time for its Revision. Am J Respir Crit Care Med [online ahead of print] 1 Aug 2022; www.atsjournals.org/doi/10.1164/rccm.202204- 0671PP

4. Therefore, in this paper we aim to analyze the role of alarmins, in particular of 28 HMGB1 and TSLP, in asthma and COPD, identifying similarities and differences which 29 may be useful for therapeutic approaches as targeted as possible. Please improve the description of study aim and underline the novelty of the study

5.  Therefore, the data we analyzed suggest a potential role of these alarm-  ins in the classification of severity of respiratory chronic diseases and their possible use as  a predictive marker of therapeutic efficacy and disease severity (Figure 1). Please, Improve this paragraph and underline the clinical implications of this observation. 

6. Figure 1. Implications of airway epithelial cell alarmins in the pathogenesis of chronic inflamma- 1 tory lung diseases. Please, improve the description of study aim. 

Author Response

(The authors gave the same response as above.)

Round 2

Reviewer 1 Report

I am happy with the changes.